# Downregulated RBM5 Enhances CARM1 Expression and Activates the PRKACA/GSK3β Signaling Pathway through Alternative Splicing-Coupled Nonsense-Mediated Decay

**DOI:** 10.3390/cancers16010139

**Published:** 2023-12-27

**Authors:** Yanping Zhang, Fang Li, Zhenwei Han, Zhihai Teng, Chenggen Jin, Hao Yuan, Sihao Zhang, Kexin Sun, Yaxuan Wang

**Affiliations:** 1Department of Urology, The Second Hospital of Hebei Medical University, Shijiazhuang 050011, China; zhangyanping@hebmu.edu.cn (Y.Z.); hanzhenwei@hebmu.edu.cn (Z.H.); tengzhihai1988@126.com (Z.T.); 29005698@hebmu.edu.cn (C.J.); yuanhao_urology@126.com (H.Y.); sunkexin1105@gmail.com (K.S.); 2Department of Cardiology, The Second Hospital of Hebei Medical University, Shijiazhuang 050011, China

**Keywords:** RBM5, CARM1, alternative splicing, nonsense-mediated mRNA decay, bladder cancer

## Abstract

**Simple Summary:**

Previous studies have demonstrated that downregulated RBM5 promotes the progression of bladder cancer. Alternative splicing (AS) plays a crucial role in the progression of cancer by promoting nonsense-mediated mRNA decay (NMD). However, whether RBM5 modulates the progression of BC through AS-NMD remains unexplored. The present study revealed that RBM5 negatively regulates the expression of CARM1 by binding directly to its mRNA and participating in the NMD process of CARM1 mRNA in BC cells. CARM1 mediates the activation of Wnt/β-catenin and RBM5 by promoting the phosphorylation of GSK3β. Protein kinase catalytic subunit alpha (PRKACA) acts as a phosphorylated kinase of GSK3β and is regulated by CARM1 at the transcription level. The results proved that there exists a regulatory mechanism for Wnt/β-catenin activation through the RBM5/CARM1/PRKACA axis and identified a new potential target for treating BC.

**Abstract:**

Downregulated RNA-binding motif protein 5 (RBM5) promotes the development and progression of various tumors, including bladder cancer (BC). Alternative splicing (AS) plays a crucial role in the progression of cancer by producing protein isomers with different functions or by promoting nonsense-mediated mRNA decay (NMD). However, whether RBM5 modulates the progression of BC through AS-NMD remains unexplored. In this study, we revealed that the downregulation of RBM5 expression promoted the expression of coactivator-associated arginine methyltransferase 1 (CARM1) in BC cells and tissues. Increased expression of CARM1 facilitated the activation of the Wnt/β-catenin axis and cell proliferation, which then contributed to the poor prognosis of patients with BC. Interestingly, RBM5 bound directly to CARM1 mRNA and participated in AS-NMD, downregulating the expression of CARM1. In addition, we revealed that protein kinase catalytic subunit alpha (PRKACA) functioned as a phosphorylated kinase of GSK3β, was regulated by CARM1 at the transcription level, and promoted the growth and progression of BC cells. Furthermore, in this study, we demonstrated a regulatory mechanism of Wnt/β-catenin activation through the RBM5/CARM1/PRKACA axis and identified a novel potential target for treating BC.

## 1. Introduction

Bladder cancer (BC) is the most common tumor of the urinary system and a prevalent cancer worldwide [1]. According to statistical data, approximately 620,000 individuals are diagnosed with BC each year and over 300,000 individuals succumb to BC annually [2]. Approximately 70% of patients who are diagnosed with BC present with non-muscle-invasive BC (NMIBC), and the remaining 30% of patients with BC already have muscle-invasive BC (MIBC) [3]. Additionally, approximately 75% of high-grade NMIBC cases progress to an aggressive state because of the distant metastases and progressiveness, causing the 5-year survival rate of patients with BC to decrease to <20% [4]. Therefore, an in-depth investigation into the molecular mechanism of BC tumorigenesis is urgently required.

The RNA-binding motif protein 5 (RBM5) was first discovered in lung cancer and was identified as a tumor suppressor because its coding region (3p21.3) is often absent in lung cancer [5]. Studies have confirmed that RBM5 is downregulated and plays an important regulatory role in various tumors, including breast cancer, prostate cancer, human vestibular schwannoma, and primary lung cancer [6,7,8]. Our previous studies revealed that downregulated RBM5 promotes the progression of BC by activating the Wnt/β-catenin signaling pathway [9]. Consistent with these findings, RBM5 has been shown to promote apoptosis through the inhibition of Wnt/β-catenin signaling [6,10]. However, the specific mechanism of action has not been fully elucidated. Some studies have shown that RBM5 regulates gene expression through alternative splicing (AS) of the gene precursor mRNA [11,12]. Moreover, RMB5 can participate in the regulation of RBM10 expression through AS coupled with nonsense-mediated mRNA decay (AS-NMD) [13]. NMD is an intracellular quality control system that prevents the intracellular accumulation of dysfunctional RNA and proteins by degrading mRNAs that contain premature termination codons or improperly spliced mRNAs [14]. However, it is unclear whether RBM5 regulates BC cell proliferation and tumor progression via AS-NMD.

Coactivator-associated arginine methyltransferase 1 (CARM1), which is a member of the arginine methyltransferase (PRMT) family, catalyzes the methylation of the protein arginine residues of guanidine nitrogen [15]. Studies have reported that CARM1 is involved in multiple cellular processes, such as DNA packaging, transcriptional regulation, mRNA splicing, and mRNA stabilization [16]. CARM1, along with the EP300/P300 and p160 families, is recruited as a promoter to activate transcription through chromatin remodeling [17]. CARM1 enhances the stability of target mRNA by methylating the arginine residues in RNA-binding proteins such as ELAVL1 and ELAV4 [18]. In addition, CARM1 is closely related to the Wnt/β-catenin signaling pathway [19,20]. Emerging research has shown that CARM1 promotes the proliferation of human osteosarcoma cells through the p-GSK3β/β-catenin/cyclinD1 signaling pathway [21]. In addition, CARM1 can regulate downstream gene expression with β-catenin/androgen receptor (AR) as a co-transcription factor [22]. Studies have also revealed that CARM1 binds to β-catenin and acts synergistically with β-catenin and p300 as a coactivator of AR [23]. However, the expression and function of CARM1 in BC remain to be fully understood.

In the present study, we revealed that RBM5 negatively regulates the expression of CARM1 by binding directly to its mRNA and participating in the NMD process of CARM1 mRNA in BC cells. CARM1 mediates the activation of Wnt/β-catenin and RBM5 by promoting the phosphorylation of GSK3β. We revealed that protein kinase catalytic subunit alpha (PRKACA) acts as a phosphorylated kinase of GSK3β and is regulated by CARM1 at the transcription level. We demonstrated a regulatory mechanism for Wnt/β-catenin activation through the RBM5/CARM1/PRKACA axis and identified a new potential target for treating BC.

## 2. Materials and Methods

### 2.1. Clinical Samples

Tissue from patients with BC and the corresponding normal bladder tissues were collected from the Department of Urology in the Second Hospital of Hebei Medical University, Shijiazhuang, Hebei Province. Part of the collected clinical tissue was fixed in 4% formaldehyde, and the remainder was stored in liquid nitrogen at −80 °C. Written informed consent was obtained from all patients from whom the clinical samples were collected. The research protocol was reviewed and approved by the Ethics Committee of the Second Hospital of Hebei Medical University.

### 2.2. Cell Culture and Treatment

Our previous results demonstrated that RBM5 was low in T24 cells and relatively high in J82 cell lines [9]. Therefore, in this study, we still used the overexpression of RBM5 in the T24 cell line and the knockdown of RBM5 in the J82 cell line. The human BC cell lines T24 and J82 were purchased from the American Type Culture Collection (Manassas, FL, USA) and stored in our laboratory [9]. The cells were cultured in minimum essential media (PM150410) and supplemented with 10% fetal bovine serum (FBS) (164210-50) and 1% P/S (PB180120) at 37 °C in a humid environment containing 5% CO_2_. Lentiviruses LV-oeRBM5, LV-shRBM5, LV-oeCARM1, LV-shCARM1, and LV-PRKACA used in this study were constructed by Shijiazhuang Biocaring Biotechnology Co., LTD., Shijiazhuang, China. The shRNA lentiviruses are all based on the LV-2N (pGLVU6/Puro) skeleton vector (Catalog No.: BCR-shRNA-LV-2N). Purinomycin was used for stable clone screening.

### 2.3. Cell Counting Kit (CCK)-8 Assay

Cell viability was evaluated using the CCK-8 assay according to the manufacturer’s instructions (MCE, HY-K0301) as previously described [24]. The cells were inoculated into 96-well microplates containing 100 μL of cells at a density of 4 × 10^3^/well (Corning Corporation, Shanghai, China). The cells were then infected with the indicated lentivirus or reagents for 48 h. Then, 10 μL of CCK-8 reagent was added to each well, and the mixture was incubated for 1–4 h. The absorbance was measured at 450 nm using a Thermo Fisher plate reader, considering the cell-free wells as blank. All experiments were performed in triplicate.

### 2.4. Xenograft Tumor Animal Model

A model of the formation of subcutaneous xenograft tumors was generated using nude mice as previously described [9,25]. In brief, male BALB/c nude mice aged 4–6 weeks and weighing 15–20 g were purchased from Standard Group Laboratory Animal Technology Ltd. (Qingdao, China). Animals were kept in the laboratory for one week to acclimate to the environment. The cultured T24 cells were infected with LV-shCARM1 or LV-shPRKACA lentivirus and enriched with puromycin. The cell lines with stable knockdown of CARM1, PRKACA, or CARM1 and PRKACA were digested with pancreatic enzymes and collected after culture expansion. Serum-free medium was used to resuspend the cells (5 × 10^6^/0.1 mL) and combined with Matrigel (BD, #356234) at a 1:1 ratio. A 0.2 mL cell suspension was injected under the dermis on the right side of the back. After tumor growth, the length and width of the mouse tumors were measured with calipers every three days until day 21. The tumor volume was calculated using the following formula: tumor volume = (length × width 2)/2. At the end of the experiment, the mice were sacrificed via cervical dislocation. The tumor samples were then dissected and collected for further examination. The animal experiment was reviewed and approved by the Ethics Committee of the Second Hospital of Hebei Medical University.

### 2.5. RNA Isolation and RT–qPCR Detection

Total RNA was extracted from the clinical tissues and cultured cells using the manual RNeasy Mini elution kit (QIAGEN, Hilden, Germany), as previously described [9,26]. The RNA concentration and mass were measured using a NanoDrop 2000 spectrophotometer. We synthesized the first strand of cDNA using a reverse transcription kit (MR05201M, Mona, Suzhou, China). The chemoHS qPCR Mix (MQ00401S, Mona) was used in the CFX96 Touch™ PCR reaction instrument to detect gene expression with GAPDH as the internal reference gene. The relative expression of the transcript was calculated using the 2^−ΔΔCt^ method. The primers are listed in Appendix A.

### 2.6. Luciferase Assay

The PRKACA 2000 bp promoter was amplified and inserted into the pGL3-basic plasmid using *MluI* and *HindIII* restriction enzymes. The sequences were confirmed via Sanger sequencing as previously described [9,26]. The 293A cells were inoculated into 24-well plates and transfected at a cell confluence of approximately 60–80%. Viruses such as LV-shAR or LV-shCARM1 were added after the PRKACA promoter was cotransfected with TK plasmid for 6 h. The cells were collected after lysis and treated with the dual luciferase assay system (Promega), and a Flash and Glow reader (LB955; Bad Wildbad, Germany) was used to detect the fluorescence values. The specific target activity was expressed using the relative activity ratio of firefly luciferase to Renilla luciferase.

### 2.7. Western Blotting

Cultured cells or clinical bladder tissues were lysed and quantified using the modified Bradford method. Then, western blotting was performed to analyze the relative protein expression levels [24]. Briefly, an equal amount of protein was loaded onto an SDS–PAGE gel and separated electrophoretically. The proteins on the gel were then transferred onto a polyvinylidene fluoride (PVDF) membrane (Millipore). The PVDF membrane was blocked with skim milk powder and incubated overnight with the following antibodies: β-actin (1:1000, sc-47778), CARM1 (1:1000, ab245466), CDK4 (1:1000, ab108357), phospho-GSK3 beta (S9) (1:500, ab75814), GSK3 beta (1:1000, ab93926), β-catenin (1:1000, ab22656), and PRKACA (1:500, ab32376). The following day, the membrane was incubated with a horseradish peroxidase (HRP)-labeled secondary antibody (1:10,000 Rockland). It was then treated with a chemiluminescent HRP substrate (Millipore), and the proteins were detected via enhanced chemiluminescence (ECL) Fuazon Fx (Vilber Bio Imaging, Marne-la-Vallée, France).

### 2.8. Minigene Splicing Reporter Assay

To evaluate the splicing of target exons after the overexpression (OE) or knockdown (KD) of *RBM5*, we cotransfected small gene splicing reporter genes into HEK293A cells. First, the cells were transfected with the Minigene plasmid and siUPF1 RNA for 6 h. LV-oeRBM5 and LV-shRBM5 as well as the corresponding control lentivirus were then transfected for 48 h. The total RNA was separated using an RNA extraction kit, and cDNA was synthesized via reverse transcription. The reverse-transcription-specific primers included forward primers (T7-F), reverse primers (BGH-R), and target-sequence-specific forward primers (CARM-E9M-F). Finally, primers targeting the CARM1 coding sequence (CARM1-coding sequence (CDS)) were used to evaluate the OE and KD efficiency via RT–PCR, using GAPDH as the internal reference gene [13].

### 2.9. Histomorphological Analysis

Clinical tissues were fixed with formalin and sliced into 5 μm thick sections. The tissue sections were stained using hematoxylin and eosin and immunohistochemically stained as previously described [6]. The tissue sections were dewaxed and antigen rescue was performed. After treatment with hydrogen peroxidase, the sections were blocked with FBS for 30 min and incubated at 4 °C overnight with the primary antibody. The enzyme-linked antibody coupled with the secondary antibody was applied to a 3,3′-diaminobenzidine-stained slide, and the sections were analyzed.

### 2.10. Chromatin Immunoprecipitation (ChIP) Assay

The ChIP assay was performed according to the manufacturer’s instructions [26] (#17-295, Sigma-Aldrich, Temecula, CA, USA). In short, T24 cells were treated with 1% formaldehyde to cross-link the protein with the DNA. The cross-linked chromatin was prepared and ultrasound-treated to an average size of 400–600 bp. The sample was diluted 10-fold and then pretreated with protein A-agarose/salmon sperm DNA at 4 °C for 30 min. The DNA fragments were immunoprecipitated overnight at 4 °C with anti-AR, anti-CARM1, or anti-IgG antibodies. After cross-linking reversal, RT-qPCR was used to detect AR or CARM1 binding on PRKACA promoters.

### 2.11. Statistical Analysis

All data were derived from the three independent experiments and expressed as the mean ± standard error of the mean. GraphPad Prism 8.0 was used for the mapping and statistical analysis. The independent student *t*-test was used to compare the means of continuous variables. *p* < 0.05 was considered statistically significant.

## 3. Results

### 3.1. RBM5 Negatively Regulates CARM1 Expression in BC

Our previous study confirmed that RBM5 was involved in the tumorigenesis of BC by regulating the Wnt/β-catenin axis [9]. To further investigate the downstream genes of RBM5 that contributed to the activation of the Wnt/β-catenin axis, we performed transcriptome sequencing and found that RBM5 OE in T24 cells modulated the expression of multiple genes (Figure 1A, Appendix A). To confirm which genes are regulated by RBM5, we selected genes with significant transcriptomic differences and verified them via RT–qPCR. The results showed that TBC1D3G, LLPHP3, and KISS1R were upregulated while PROX2 and CARM1 were downregulated in RBM5 OE T24 cells (Figure 1B). In contrast, TBC1D3G and CARM1 were elevated while RADIL and MANSC4 were reduced in RBM5 KD J82 cells (Figure 1C). Only CARM1 was reduced in RBM5-OE and elevated in RBM5-KD cells. Subsequently, we measured the CARM1 protein expression and confirmed that RBM5 negatively regulated CARM1 in BC cells (Figure 1D–G; Appendix A). We then performed morphological tests on the collected clinical samples. The immunohistochemical staining results showed that the expression of CARM1 was significantly upregulated in the BC tissues (Figure 1H,I). Western blotting and RT–qPCR results confirmed the high expression of CARM1 in the BC tissues (Figure 1J,K; Appendix A). Furthermore, data analysis from The Cancer Genome Atlas (TCGA) revealed the same result (Figure 1L). The Kaplan–Meier analysis from the TCGA database revealed that the higher expression of CARM1 contributes to the poor outcome of patients with BC (Figure 1M). These findings suggest that CARM1, a negatively regulated gene of RBM5, is upregulated in BC tissues and may play a role in the progression of BC.

### 3.2. CARM1 Plays a Key Role in the Proliferation of BC Cells

To further investigate the functions of CARM1 in BC cells, we performed the following gain-and-loss experiment. First, we constructed the CARM1 OE lentivirus LV-oeCARM1 and two CARM1 KD lentiviruses—LV-shCARM1-1 and LV-shCARM1-2—and then transfected them into BC cells. Western blotting and RT–qPCR results showed that transfection with LV-oeCARM1 significantly increased the expression of CARM1 and CDK4 in J82 cells compared with the negative control. In contrast, T24 cells transfected with LV-shCARM1 suppressed the expression of CARM1 and CDK4 at mRNA and protein levels (Figure 2A–C; Appendix A). The cells were transfected with LV-shCARM1 or LV-oeCARM1, after which the CCK8 assay was used to detect cell viability. As shown in Figure 2D, *CARM1* KD in T24 cells reduced their viability, whereas *CARM1* OE in J82 cells promoted the proliferation of cells compared with the negative control. To clarify whether CARM1 mediates the regulation of proliferation via RBM5, we performed rescue experiments. First, we infected T24 cells with LV-shCARM1 or LV-shRBM5 or coinfected them together. Then, the colony formation assay was used to detect cell proliferation. The results revealed that *CARM1* KD in T24 cells significantly reduced cell proliferation; however, this induction effect by CARM1 could be partially reversed by *RBM5* KD (Figure 2E,F). Similarly, *RBM5* OE further enhanced the inhibitory effect of *CARM1* KD (Figure 2G,H). Taken together, these results suggest that CARM1 contributes to the regulation of the proliferation of BC cells via RBM5.

### 3.3. CARM1 Promotes the Activation of the Wnt/β-Catenin Axis in BC Cells

To clarify whether CARM1 is involved in the activation of the Wnt/β-catenin axis in BC cells, we performed the following gain-and-loss experiments. As shown in Figure 3A, transfection with the *CARM1* OE lentivirus significantly increased the expression of β-catenin and p-GSK3β compared with the negative control (Figure 3A,B; Appendix A). However, *CARM1* KD led to a decrease in the expression of β-catenin and p-GSK3β in T24 cells (Figure 3C,D; Appendix A), indicating that CARM1 can affect GSK3β phosphorylation and Wnt/β-catenin activity. Then, we determined whether CARM1 mediates the relationship between RBM5 and the activation of the Wnt/β-catenin axis. The T24 cells were transfected with shCARM1 or shRMB5 or were cotransfected with both of them. Western blotting demonstrated that *RBM5* KD increased the protein levels of p-GSK3 and β-catenin; however, this increase could be partially offset by transfection with LV-shCARM1 together (Figure 3E,F; Appendix A). In addition, cells were transfected with LV-shCARM1 and treated with or without XAV930, an inhibitor of the Wnt/β-catenin pathway. Western blotting revealed that XAV930 effectively reduced the activation of the Wnt/β-catenin pathway; however, this reduction effect was partially counteracted by the upregulation of *CARM1* expression in BC cells (Figure 3G,H; Appendix A). These data demonstrate that CARM1 plays a key role in the RMB5-induced activation of the Wnt/β-catenin axis.

### 3.4. RBM5 Negatively Regulates the Expression of CARM1 via AS-NMD

Previous studies have shown that RBM5 regulates gene expression by participating in AS-NMD [13]. As an RNA-binding protein, we first examined whether RBM5 binds to CARM1 RNA. As shown in Figure 4A,B, the RIP assay revealed that RBM5 can bind to and enrich the CDS motif of the CARM1 RNA. We overexpressed *RBM5* and administered doxycycline (Dox) to inhibit RNA synthesis in the cell; subsequently, we detected CARM1 expression and CARM1 splice variants. The results revealed that Dox treatment reduced CARM1-CDS while increasing the RNA content of CARM1 lacking exon 9 (Figure 4C,D). We obtained the same results at the protein level (Figure 4E,F; Appendix A). In addition, using Minigene splicing reporters, we found that *RBM5* OE inhibited the expression of CARM1-CDS and increased the expression of CARM1 splice variants that lack exon 9. *RBM5* KD increased the CARM1-CDS content and reduced the skipping of exon 9 (Figure 4G–I). The expression levels of CARM1 splice variants that lack exon 9 significantly increased following NMD inhibition via UPF1 depletion (Figure 4J,K). Moreover, the skipping of exon 9 increased further following NMD inhibition through cycloheximide treatment (Figure 4L,M). Altogether, these findings indicate that RBM5 negatively regulates the mRNA and protein expression of CARM1 via AS-NMD.

### 3.5. PRKACA Is Positively Regulated by CARM1 in BC Cells

Previous studies have reported that GSK3β phosphorylation is often regulated by phosphorylated kinases such as PKC, PKA, and AKT [27]. To determine which phosphokinases are involved in CARM1-mediated GSK3β phosphorylation, we overexpressed or knocked-down CARM1 in BC cells and detected the expression of phosphokinases. The RT–qPCR results revealed that only PRKACA was positively regulated by CARM1 in BC cells (Figure 5A,B). Then, western blotting revealed that the upregulation of CARM1 promoted while the downregulation of CARM1 reduced the PRKACA protein level in BC cells compared with the negative control (Figure 5C–F; Appendix A). Next, we measured PRKACA expression in BC tissues. RT–qPCR and western blotting revealed that the mRNA and protein expression of PRKACA was significantly increased in BC cells compared with that in normal bladder tissue (Figure 5G,H; Appendix A). High CARM1 expression was positively correlated with PRKACA in BC (Figure 5I). Moreover, the prognostic analysis from the TCGA database showed that the high expression of PRKACA predicted a poor prognosis in patients with BC (Figure 5J). We then performed rescue experiments to identify whether PRKACA was involved in the regulation of cell proliferation by CARM1 in BC cells using the CCK8 and colony formation assays. The CCK-8 results showed that the depletion of PRKACA significantly reduced cell viability, while the concomitant CARM1 OE partially reversed these effects by silencing PRKACA. However, deletion of both CARM1 and PRKACA further enhanced the inhibitory effect of CARM1-silencing alone (Figure 5K,L). The colony formation assay showed that the upregulation of CARM1 promoted cell proliferation, while this effect was partly abolished by the cotransfection (Figure 5M,N). Taken together, these results demonstrated that the high level of PRKACA, which was positively regulated by CARM1, promoted the proliferation of BC cells.

### 3.6. PRKACA Is Involved in CARM1-Promoted the Phosphorylation of GSK3β

Emerging studies have suggested that CARM1 is a transcriptional coactivator of AR-mediated signaling [22]. We aimed to determine whether CARM1 regulates PRKACA expression as a transcriptional coactivator of AR. Briefly, BC cells were transfected with oeCARM1 and shAR lentivirus alone or together, and RT–qPCR results showed that *CARM1* overexpression promoted the partially offset PRKACA mRNA expression while depleting AR simultaneously (Figure 6A). Next, we performed the luciferase assay to determine whether CARM1 and AR could coregulate PCKACA promoter activity. As shown in Figure 6B, the depletion of AR obviously decreased the luciferase activity of the PCKACA promoter, and the simultaneous *CARM1* OE partially reversed these inhibitory effects. In addition, sequence analysis revealed that there are several AR binding sites in the PCKACA promoter (Figure 6C). We determined whether AR and CARM1 regulate PCKACA expression by binding to these sites. The results of ChIP–PCR showed that both AR and CARM1 could bind to the proximal region (−1435 to −1211 and −989 to −751) of the promoter. These findings suggest that CARM1 regulates PCKACA expression in a transcriptional manner.

To elucidate whether PRKACA is a phosphate kinase of GSK3β in BC cells, we performed *PRKACA* OE or KD in BC cells. Western blotting revealed that the upregulation of PRKACA expression markedly increased the phosphorylation of GSK3β and β-catenin. In contrast, the downregulation of PRKACA expression in T24 cells suppressed the phosphorylation of GSK3β and β-catenin compared with the negative control (Figure 6E,F; Appendix A). Furthermore, the rescue experiment revealed that *CARM1* OE upregulated the phosphorylation of GSK3β and β-catenin; however, this effect could be reversed by cotransfection with shPRKACA (Figure 6G,H; Appendix A). Similarly, we infected T24 cells with LV-shRBM5 or LV-shPRKACA or cotransfected them together. Western blotting revealed that the downregulation of RMB5 expression induced the phosphorylation of GSK3β and β-catenin; however, this induction by RBM5 could be reversed by simultaneous *PRKACA* KD (Figure 6I,J; Appendix A). Finally, T24 cells were transfected with overexpressed PRKACA and treated with XVA939, an inhibitor of the Wnt/β-catenin pathway. Western blotting showed that regardless of PRKACA overexpression, blocking the Wnt/β-catenin pathway inhibited the promotion of p-GSK3β and β-catenin expression by PRKACA (Figure 6K,L; Appendix A). These results suggest that as the phosphate kinase of GSK3β, PRKACA plays a key role in the activation of Wnt/β-catenin via the RBM5/CARM1 axis in BC cells.

### 3.7. Blocking the RBM5/CARM1/PRKACA Axis Reduces BC Cell Proliferation In Vivo

To clarify the role of RBM5/CARM1/PRKACA in the proliferation of BC cells in vivo, we established a nude mouse model. First, T24 cells with stably depleted CARM1 or PRKACA alone or in combination transfected with the corresponding lentivirus, and the negative control T24 cells were implanted into the nude mice. Tumor volumes and weight were measured after 21 days. As shown in Figure 7A, CARM1 or PRKACA KD led to a significantly smaller volume and lighter weight of tumor tissues compared with the negative control. However, the combined depletion of both CARM1 and PRKACA had the smallest volume and lightest weight (Figure 7B,C). Next, we detected the CDK4, β-catenin, and CARM1 protein levels in the xenograft tumors. Immunohistochemical staining showed that the deletion of CARM1 or PRKACA alone downregulated the CDK4 and β-catenin protein levels. The combined deletion of both CARM1 and PRKACA further decreased the expression levels of these proteins (Figure 7D). Western blotting revealed the same results, and the expression of p-GSK3β was significantly inhibited (Figure 7E,F; Appendix A). These findings demonstrated that blocking the RBM5/CARM1/PRKACA pathway inhibited the in vivo cell proliferation and tumor progression (Figure 8).

## 4. Discussion

RNA-binding proteins (RBPs) are a family of proteins that bind directly to single- or double-stranded RNA and are involved in the regulation of various RNA processes [28]. RBPs play a critical role in gene regulation by participating in several processes of RNA metabolism, such as RNA splicing, mRNA stabilization and degradation, and RNA localization and transfer [29]. Previous studies have demonstrated that RBPs play essential roles in the tumorigenesis of BC cells [30,31,32]. For example, FXR1 functions as an oncogene in BC by binding to tumor necrosis factor receptor-associated factor 1 (TRAF1) mRNA, which leads to mRNA stabilization [31]. Moreover, BRCC3 was involved in BC cell proliferation and invasion by binding to TRAF1, thus influencing the activation of the NF-κB pathway [33]. In addition, the analysis from the TCGA database revealed that the mutated or aberrant expression of RBPs was closely related to the prognosis in patients with BC [34]. Our previous study revealed that RBM5, as an important RBP member, was downregulated in BC tissues and cells [9]. RBM5 OE effectively reduced the activation of the Wnt/β-catenin axis, which resulted in a decrease in cell proliferation [6]. In this study, we confirmed that RBM5 negatively regulated the expression of CARM1 by directly binding to its mRNA and participating in the NMD process of CARM1 mRNA in BC cells. CARM1 promoted the expression of PRKACA, a phosphate kinase, at the transcriptional level. PRKACA mediated RBM5/CARM1 and induced the activation of Wnt/β-catenin in BC cells.

AS is an important cause of eukaryotic transcriptome and proteome diversity [35]. AS plays a key role in disease progression by producing different functional protein isomers [36], noncoding RNAs [37], and NMD substrates [38]. The molecular mechanisms by which eukaryotes acquire numerous splicing regulators have been explored, but the mechanism by which they control their own expression to produce specific RNA splicing patterns under different physiological conditions is not fully understood. NMD is an important intracellular quality control system for mRNA surveillance [39]. NMD prevents the intracellular accumulation of dysfunctional RNAs and proteins by degrading mRNAs that contain premature termination codons or improperly spliced mRNAs. Moreover, NMD is an essential mRNA degradation pathway. Therefore, the downregulated expression of RBM5 may increase mRNA and protein levels of CARM1 by inhibiting NMD. Some studies have shown that RBM5 could regulate gene expression through the AS of pre-mRNA genes [40]. Importantly, RBM5, an analog of RBM10, regulates downstream gene expression through AS-NMD [13]. However, few studies have considered RBM5 itself as a key factor that regulates the downstream gene expression through AS-NMD. In the present study, we revealed that RBM5 negatively regulates the mRNA and protein expression of CARM1 via AS-NMD.

The human cAMP-dependent PRKACA gene is located on the p13.1 antichain of chromosome 19 [41]. PRKACA has 10 exons that are transcribed into 351 amino acid proteins (40 kDa) [42,43]. PRKACA is a catalytic subunit of PKA, which is mainly responsible for the phosphorylation of downstream proteins and substrates to alter their activity [44]. An increasing number of studies have shown that PRKACA may promote the progression of cancer through kinases, such as PKA catalytic subunits, which are detected in the extracellular serum in various cancers, such as lymphoma as well as colon, kidney, rectal, prostate, lung, and adrenal cancers [44]. An increase in the number of PRKACA transcripts was also observed in hepatocellular carcinoma and breast cancer [45,46]. In addition, transcripts of the DNAJB1–PRKACA fusion in fibrous lamellar hepatocellular carcinoma cause a fatal disease that has no specific treatment [47]. A large number of studies have reported that PKA promotes the activation of the Wnt/β-catenin signaling pathway by regulating GSK3β phosphorylation [48,49]. However, its role in BC remains unclear. In this study, we revealed that PRKACA is upregulated in BC tissues and mediates RBM5 to regulate the phosphorylation of Wnt/β-catenin and progression of BC. The limitations of this study are as follows: (1) More clinical samples and more reliable animal experiments, such as patient-derived xenografts (PDX), are needed to enhance the validity of the results. (2) Whether the relationship between AR and RBM5 has a feedback loop regulation mode should be investigated. The abovementioned problems warrant further in-depth studies.

## 5. Conclusions

In summary, RBM5 regulates the expression of CARM1 in BC cells through the mRNA AS-NMD process. CARM1 is involved in the RBM5 regulation of Wnt/β-catenin activation by promoting GSK3β phosphorylation. PRKACA is a phosphorylated kinase of GSK3β and is regulated by CARM1 at the transcriptional level. These findings reveal the regulatory mechanisms by which the RBM5/CARM1/PRKACA axis activates the Wnt/β-catenin pathway, identifying potential new targets for treating BC.

## Figures and Tables

**Figure 1 cancers-16-00139-f001:**
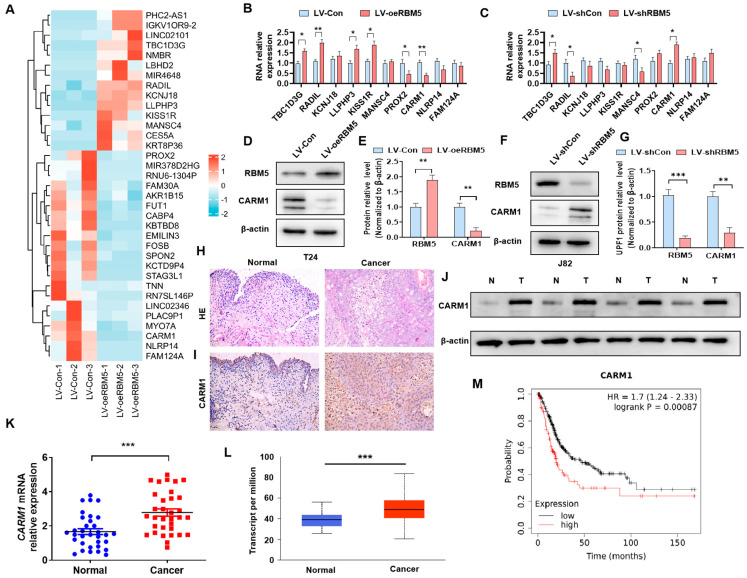
CARM1 is a downstream gene of RBM5 and is upregulated in bladder cancer (BC) tissues. (**A**) T24 cells were transfected with the LV−oeRBM5 lentivirus, and the gene expression profile was used to target the downstream genes of RBM5. The heat map shows the upregulated genes in red and the downregulated genes in blue. (**B**,**C**) T24 cells were transfected with LV−oeRBM5 and LV−Con, whereas J82 cells were transfected with LV−shRBM5 and LV−shCon. Subsequently, RT−qPCR was used to detect the mRNA expression of candidate genes. (**D**,**E**) T24 cells were transfected with LV−oeRBM5 and LV−Con, and western blotting was used to examine the CARM1 protein expression. (**F**,**G**) J82 cells were transfected with LV−shRBM5 and LV−shcon, and CARM1 protein expression was detected via western blotting. (**H**) Hematoxylin and eosin staining was used to analyze the normal bladder tissue and BC tissue. (**I**) The CARM1 expression in the normal bladder and BC tissues was detected via immunohistochemical staining. (**J**,**K**) The expression of CARM1 protein and mRNA was evaluated in normal and BC tissues via western blotting and RT−qPCR. (**L**) CARM1 mRNA expression was analyzed based on the data from The Cancer Genome Atlas (TCGA) database. (**M**) High CARM1 expression was associated with a poor prognosis and a low overall survival rate in patients with BC from the TCGA database. All data were derived from three independent experiments and expressed as the mean ± standard error of the mean (SEM). * *p* < 0.05, ** *p* < 0.01, *** *p* < 0.001 vs. their corresponding controls.

**Figure 2 cancers-16-00139-f002:**
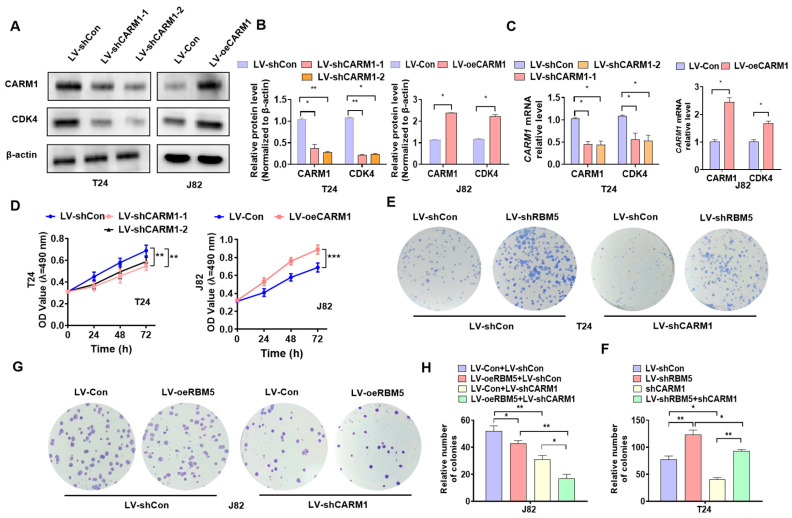
CARM1 promotes the proliferation of bladder cancer (BC) cells. (**A**–**C**) T24 cells were transfected with LV−shCARM1−1 and LV−shCARM1−2, and J82 cells were transfected with LV−oeCARM1 or their corresponding control lentivirus, and western blotting and RT–qPCR were used to detect the CARM1 and CDK4 proteins (**A**,**B**) or mRNA (**C**) expression. (**D**) A CCK−8 assay was performed to examine cell viability after the treatment described in (**A**). (**E**,**F**) T24 cells were transfected with LV−shRBM5 or LV−shCARM1 alone or in combination, after which the colony formation assay was used to measure cell proliferation. (**G**,**H**) J82 cells were transfected with LV−oeRBM5 or LV−shCARM1 alone or in combination, and the colony formation assay was used to measure cell growth. All data were derived from three independent experiments and expressed as the mean ± standard error of the mean (SEM). * *p* < 0.05, ** *p* < 0.01, *** *p* < 0.001 vs. their corresponding controls.

**Figure 3 cancers-16-00139-f003:**
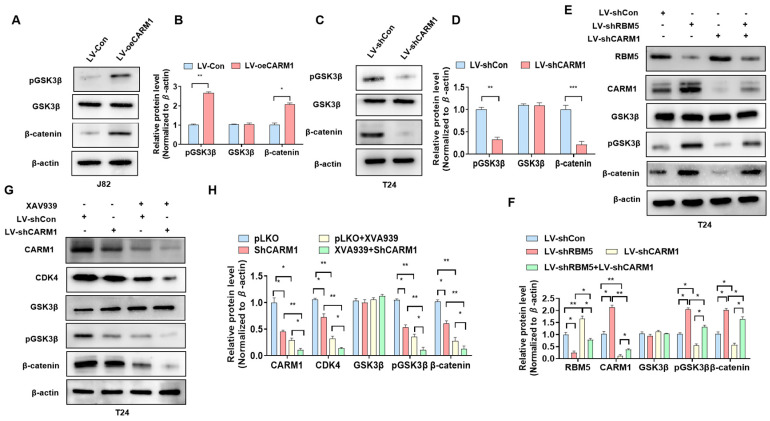
CARM1 is involved in the activation of the Wnt/β-catenin signaling pathway regulated by RBM5. (**A**) J82 cells were transfected with LV-oeCARM1 or LV-Con, and western blotting was performed to examine the protein expression of p-GSK3β, GSK3β, and β-catenin. (**B**) Quantitative analysis of (**A**). (**C**) T24 cells were transfected with LV−shCARM1 or control lentivirus, after which p-GSK3β, GSK3β, and β-catenin protein levels were assessed via western blotting. (**D**) Quantitative analysis of (**C**). (**E**) T24 cells were transfected with LV−shRBM5 or LV−shCARM1 alone or in combination; afterward, western blotting was performed to detect the protein expression of p-GSK3β, GSK3β, and β-catenin. (**F**) Quantitative analysis of (**E**). (**G**) T24 cells were transfected with LV-shCARM1 or LV-shCon and then treated with XAV939. The CDK4, p-GSK3β, GSK3β, and β-catenin protein levels were measured via western blotting. (**H**) Quantitative analysis of (**H**). All data were derived from three independent experiments and expressed as the mean ± standard error of the mean (SEM). * *p* < 0.05, ** *p* < 0.01, *** *p* < 0.001 vs. their corresponding controls.

**Figure 4 cancers-16-00139-f004:**
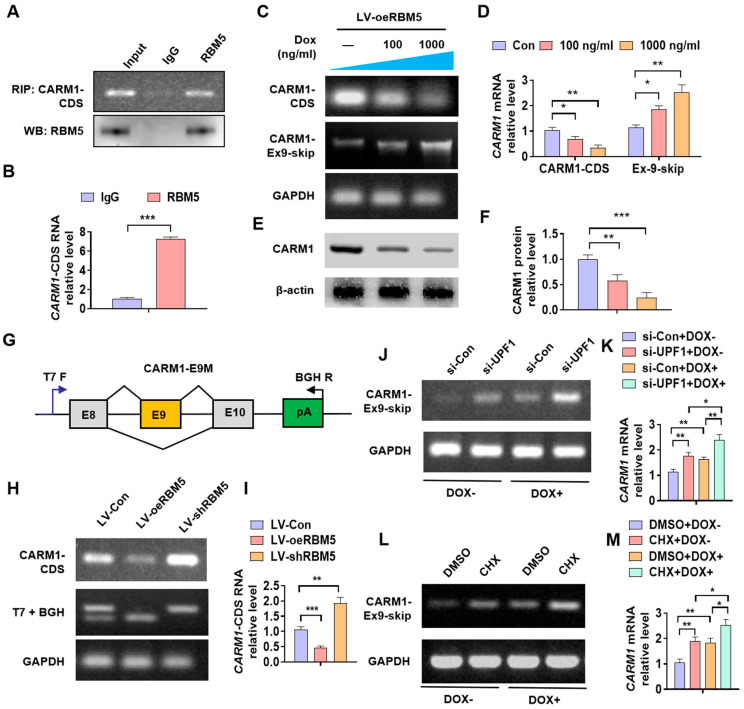
RBM5 inhibits the expression of CARM1 via alternative splicing-coupled nonsense-mediated mRNA decay (AS-NMD). (**A**) An RNA-binding protein immunoprecipitation (RIP) assay was performed using an RBM5 antibody to examine the interaction of RBM5 with CARM1 RNA. Upper panels: representative gel images; lower panels: western blot results. (**B**) The enrichment of the CARM1-coding sequence (CDS) RNA during the RIP assay was detected via RT–qPCR. (**C**,**D**) After transfection of T24 cells with LV-oeRBM5 and treatment with doxycycline (Dox), the levels of CARM1 mRNA and splice variants that lacked exon 9 (CARM1-Ex9-skip) were assessed using RT−PCR. GAPDH was considered the internal reference gene. (**E**,**F**) T24 cells were treated as described in (**C**), and western blotting was performed to examine the CARM1 protein expression. (**G**) Schematic of a Minigene splicing reporter. (**H**,**I**) RT−PCR analysis of the expression levels of CARM1 splice variants that lack exon 9 from Minigene splicing reporters (CARM1-E9M) coinfected with LV-shRBM5 and LV-oeRBM5, as described in (**B**). n = 4 biological replicates of RBM5-E6M. (**J**,**K**) RT−PCR analysis of the changes in expression of CARM1 splice variants that lack exon 9 following nonsense-mediated mRNA decay (NMD) inhibition via UPF1 depletion and Dox treatment. (**L**,**M**) CARM1 splice variants that lack exon 9 following NMD inhibition via cycloheximide (CHX) and Dox treatment. All data were derived from three independent experiments and expressed as the mean ± standard error of the mean (SEM). * *p* < 0.05, ** *p* < 0.01, *** *p* < 0.001 vs. their corresponding controls.

**Figure 5 cancers-16-00139-f005:**
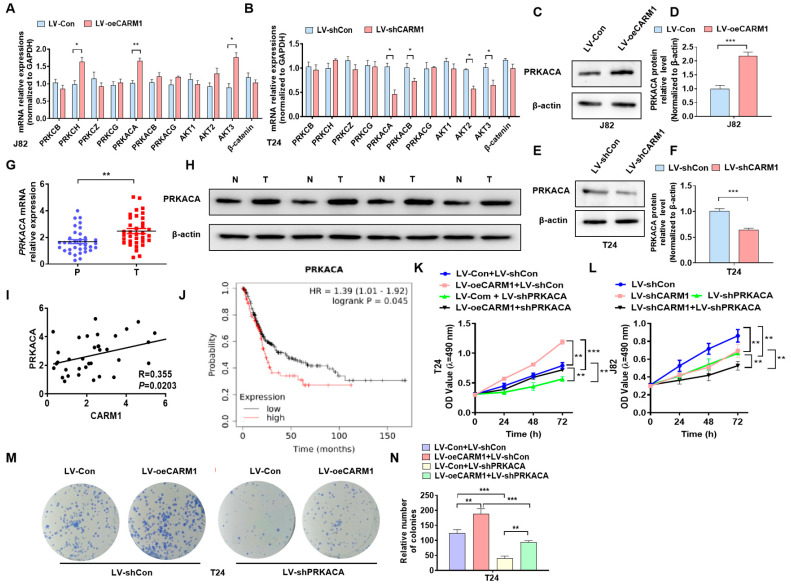
CARM1 regulates the expression of PRKACA in bladder cancer (BC). (**A**,**B**) J82 cells were transfected with LV−Con or LV−oeCARM1, or T24 cells were transfected with LV−shCon or LV−shCARM1, and RT–qPCR was used to examine the expression of the indicated pathway genes. (**C**,**D**) Western blotting was used to detect the PRKACA protein levels in J82 cells transfected with LV−Con or LV−oeCARM1. (**E**,**F**) T24 cells were transfected with LV−shCon or LV−shCARM1, and PRKACA protein expression was assessed via western blotting. (**G,H**) RT−qPCR and Western blot were used to explain PRKACA mRNA and protein expression in the clinical tissues. (**I**) Correlation analysis of PRKACA and CARM1 in bladder tissues (R = 0.355, *p* = 0.0203). (**J**) The Kaplan–Meier method was used to analyze the relationship between PRKACA expression and overall survival in patients with BC. (**K**,**L**), T24 and J82 cells were transfected with the indicated lentiviruses, and the CCK−8 assay was used to evaluate the cell viability. (**M**,**N**) T24 cells were transfected with LV−oeCARM1 or LV−shPRKACA alone or in combination, and a colony formation assay was performed to evaluate cell growth. All data were derived from three independent experiments and expressed as the mean ± standard error of the mean (SEM). * *p* < 0.05, ** *p* < 0.01, *** *p* < 0.001 vs. their corresponding controls.

**Figure 6 cancers-16-00139-f006:**
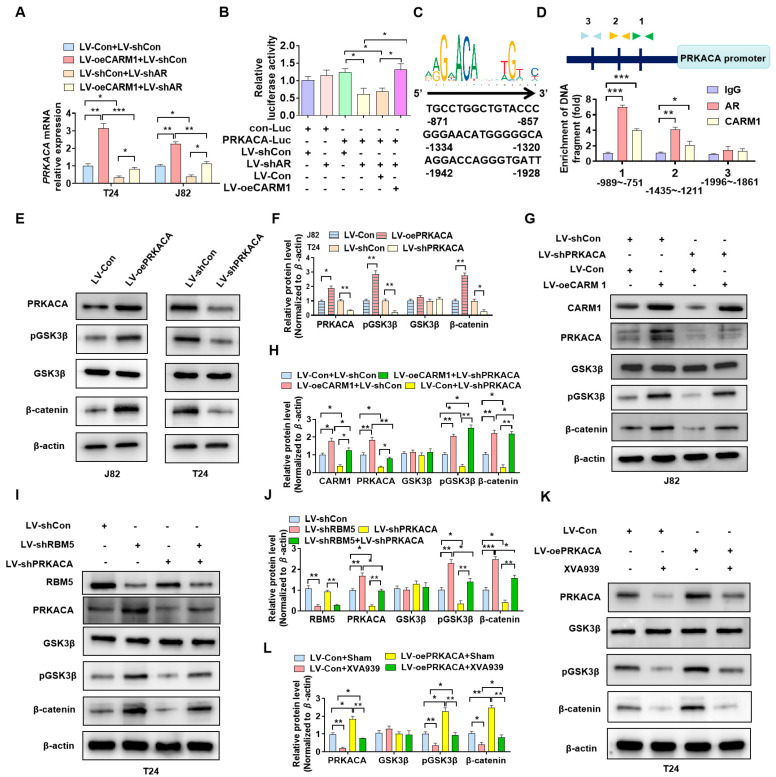
PRKACA is involved in CARM1−promoted GSK3β phosphorylation. (**A**) T24 and J82 cells were transfected with LV−oeCARM1 and LV−shAR, and RT–qPCR was used to detect PRKACA mRNA expression. (**B**) The dual luciferase reporter gene assay revealed that CARM1 and androgen receptor (AR) coregulated PRKACA promoter activity. (**C**) Analysis of the potential AR−CARM1 cobound motif on the promoter of the *PRKACA* gene. (**D**) ChIP–PCR was performed to verify the binding sites of AR and CARM1 on the PRKACA promoter using AR and CARM1 antibodies. (**E**,**F**) Cells were transfected with the indicated lentiviruses, and western blotting was performed to assess the expression of p-GSK3β, GSK3β, and β-catenin. (**G**,**H**) J82 cells were transfected with LV−shPRKACA or LV−oeCARM1 alone or in combination, and the protein levels of p-GSK3β, GSK3β, and β-catenin were measured via western blotting. (**I**,**J**) In T24 cells, *RBM5* and *PRKACA*, alone or in combination, were knocked down by lentivirus, and western blotting was performed to detect the expression of p-GSK3β, GSK3β, and β-catenin. (**K**,**L**) T24 cells were transfected with LV−PRKACA or LV−Con and then treated with XVA939, a Wnt/β-catenin pathway inhibitor. Western blotting was performed to evaluate the protein levels. All data were derived from three independent experiments and expressed as the mean ± standard error of the mean (SEM). * *p* < 0.05, ** *p* < 0.01, *** *p* < 0.001 vs. their corresponding controls.

**Figure 7 cancers-16-00139-f007:**
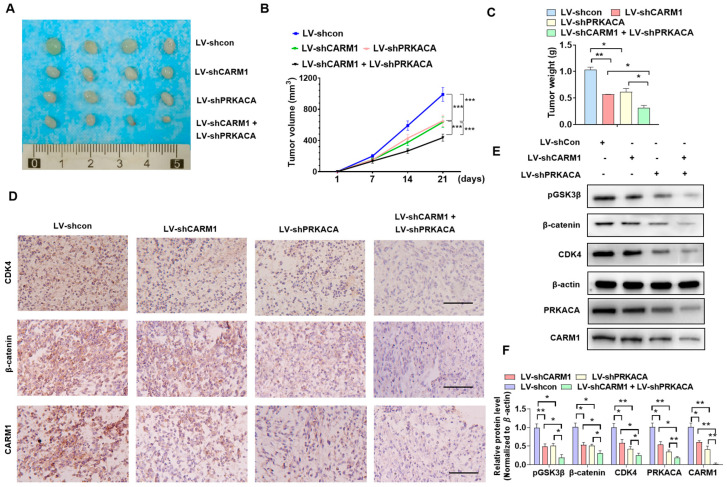
Blocking the RBM5/CARM1/PRKACA axis reduces the proliferation of in vivo bladder cancer (BC) cells. (**A**) T24 cells were stably knocked down with CARM1 or PRKACA, alone or in combination, and then implanted into nude mice for 21 days to establish xenograft tumors. The size of the xenograft tumors in each group is shown. (**B**,**C**) Xenograft tumor volumes and wet weight were determined after tumor resection. (**D**) The expression of CDK4, β-catenin, and CARM1 in the xenograft tumors was detected via immunohistochemistry. Scale bar = 100 μm. (**E**,**F**) Western blotting was used to assess the expression of p-GSK3β, CDK4, and β-catenin in the xenograft tumor tissue. All data were derived from three independent experiments and expressed as the mean ± standard error of the mean (SEM). * *p* < 0.05, ** *p* < 0.01, *** *p* < 0.001 vs. their corresponding controls.

**Figure 8 cancers-16-00139-f008:**
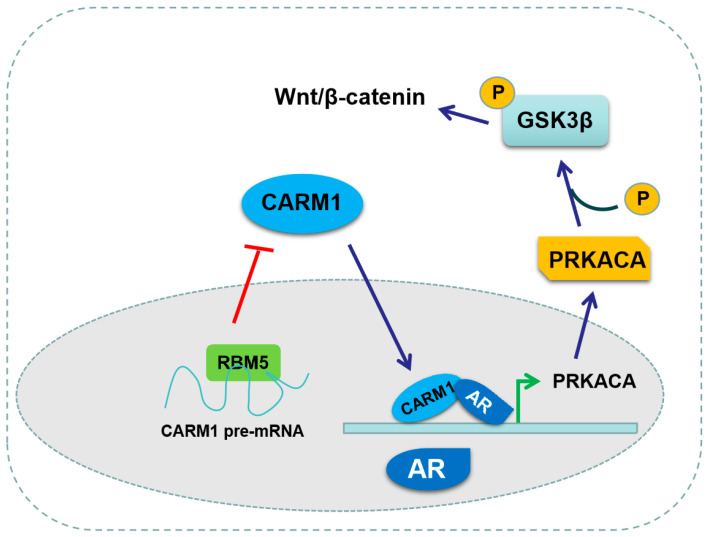
Proposed model of RBM5/CARM1/PRKACA axis regulation in the progression of bladder cancer (BC).

## Data Availability

The published article includes all data sets generated/analyzed for this study.

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
