# Peer review of "Downregulated RBM5 Enhances CARM1 Expression and Activates the PRKACA/GSK3β Signaling Pathway through Alternative Splicing-Coupled Nonsense-Mediated Decay"

_cancers, 2023, doi:10.3390/cancers16010139_

Round 1

Reviewer 1 Report

Comments and Suggestions for Authors

In this manuscript, the authors elucidate the regulatory mechanism of RNA-binding motif 5 (RBM5) in modulating the expression of coactivator-associated arginine methyltransferase 1 (CARM1). The study demonstrates that downregulation of RBM5 leads to an upregulation of CARM1 expression, subsequently activating the Wnt/β-catenin signaling pathway through protein kinase cAMP-activated catalytic subunit alpha (PRKACA). Furthermore, the authors reveal that RBM5 expression downregulates CARM1 through alternative splicing-mediated nonsense-mediated mRNA decay (AS-NMD). While carefully reviewing the manuscript, several suggestions are proposed to enhance clarity and provide a more comprehensive understanding of the presented data.

1.       The authors selected the T24 cell line for RBM5 overexpression (OE) experiments and the J82 cell line for RBM5 downregulation (KD) experiments. However, clarification is needed regarding the rationale behind using different cell lines for OE and KD experiments, especially since both cell lines express RBM5. It is suggested to consider KD experiments in the same cell line first, followed by mutant-resistant OE experiments.

2.       The manuscript lacks clear explanations in both the results and methods sections regarding the selection of different cell lines for OE and KD experiments. A revision is recommended to provide detailed descriptions, enhancing the overall understanding for readers.

3.       Although the authors mention the construction of RBM5 OE, RBM5 KD, CARM1 OE, CARM1 KD, PRKACA KD, and PRKCA OE cells, corresponding experimental data for KD or OE for that protein is not presented. The authors should include data demonstrating the overexpression or knockdown of each gene throughout the manuscript, including luciferase experiments using the LV-PRKACA construct.

4.       The manuscript does not specify whether experiments related to RBM5, CARM1, PRKACA knockdown, or OE were performed using stable clones or transient transfections. This detail should be explicitly mentioned.

5.       Experiments involving either double knockdown or knockdown of one gene and overexpression of another gene simultaneously are mentioned but not presented. It is crucial to include the data showing KD or OE in those cell lines in the manuscript.

6.       Figure 4 references the use of si-UPF1, but the knockdown of UPF1 using si-UPF1 is not shown. Authors are encouraged to present this validation data.

7.       In Figure 6B, the relative luciferase activity of two controls (bar 3 and bar 5) differs. The authors should explain this discrepancy.

8.       Although ChIP experiments are depicted in Figure 6D, the manuscript lacks a detailed description of the ChIP protocol. It is recommended to include this information for transparency.

9.       The manuscript does not mention the target sequences for the numerous shRNA constructs used. Authors should provide these sequences or, if commercially obtained, include catalog numbers for transparency.

10.   In Section 3.7, the last sentence of results refers to Fig. 8, but according to the data, it should be Fig. 7E and F. Authors are advised to correct this reference for accuracy.

Author Response

Point-by-point response to referees

 Reviewer 1#

  1. The authors selected the T24 cell line for RBM5 overexpression (OE) experiments and the J82 cell line for RBM5 downregulation (KD) experiments. However, clarification is needed regarding the rationale behind using different cell lines for OE and KD experiments, especially since both cell lines express RBM5. It is suggested to consider KD experiments in the same cell line first, followed by mutant-resistant OE experiments.

Reply: Thank you for your comments and suggestions. However, in our previous study 1, we have examined the expression of RBM5 protein in a normal bladder cell line (SV-HUC-1) and in a series of bladder cancer cell lines (T24, UM-UC-3, J82, and RT4). And the results revealed that RBM5 expression was significantly decreased in 3 tumor cell lines (T24, UM-UC-3, and RT4) when compared with the normal cells ( Figure S1 A and B). Among them, the expression of RBM5 was most significantly down-regulated in T24 cell line, but relatively high in J82 cell line. Therefore, it is unnecessary to present this part of the results in this study. As in previous publications, we overexpressed RBM5 in the T24 cell line and knocked down RBM5 in J82 in this study. 

Figure S1(FASEB J. 2019 Oct;33(10):10973-10985)

  1. The manuscript lacks clear explanations in both the results and methods sections regarding the selection of different cell lines for OE and KD experiments. A revision is recommended to provide detailed descriptions, enhancing the overall understanding for readers.

Reply: Thank you for your comments and suggestions. Our previous results demonstrated that RBM5 was low in T24 cells and relatively high in J82 cell lines. Therefore, in this study, we still used overexpression of RBM5 in T24 cell line and knockdown of RBM5 in J82 cell line. Relative comments have been added to the Method section and marked in red.

  1. Although the authors mention the construction of RBM5 OE, RBM5 KD, CARM1 OE, CARM1 KD, PRKACA KD, and PRKCA OE cells, corresponding experimental data for KD or OE for that protein is not presented. The authors should include data demonstrating the overexpression or knockdown of each gene throughout the manuscript, including luciferase experiments using the LV-PRKACA construct.

Reply: Thank you for your suggestions. The protein expression of RBM5 OE or RBM5 KD was added to Figures 1D and 1F. The protein expressions of CARM1 OE and CARM1 KD were shown in Figure 2A. In addition, the results of PRKACA KD and PRKACA OE were added in Figures 6E.

Figure 1D and 1F

Figure 2A

Figure 6E

  The luciferase experiments of this project examines the activity of PRKACA promoters. We also performed experiments, and the results showed that infection with LV-PRKACA overexpression vectors did not enhance the activity of PRKACA promoters (Figure S2). Please check!

Figure S2

  1. The manuscript does not specify whether experiments related to RBM5, CARM1, PRKACA knockdown, or OE were performed using stable clones or transient transfections. This detail should be explicitly mentioned.

Reply: Sorry for that! Lentiviruses LV-oeRBM5, LV-shRBM5, LV-oeCARM1, LV-shCARM1 and LV-PRKACA used in this study were constructed by Shijiazhuang Biocaring Biotechnology Co., LTD.  Purinomycin was used for stable clones screening. These comments have added in the Method section and marked in red. Please check!

  1. Experiments involving either double knockdown or knockdown of one gene and overexpression of another gene simultaneously are mentioned but not presented. It is crucial to include the data showing KD or OE in those cell lines in the manuscript.

Reply: Thank you for your suggestions. The expression of proteins involved in genes that are knocked out or overexpressed or doubly expressed has been added to the panel. RBM5 and CARM1 in Figure 3E; CARM1 in Figure G; CARM1 and PRKACA in Figure 6G; RBM5 and PRKACA in Figure 6I; PRKACA in Figure 6K; CARM1 and PRKACA in Figure 7E; Please check!

Figure 3E

Figure 3G

Figure 6G

Figure 6I

Figure 6K

Figure 7E

  1. Figure 4 references the use of si-UPF1, but the knockdown of UPF1 using si-UPF1 is not shown. Authors are encouraged to present this validation data.

Reply: Thank you for your suggestions. The protein expression of si-UPF1 was added in Supplementary Figure 1. Please check!

Supplementary Figure 1

  1. In Figure 6B, the relative luciferase activity of two controls (bar 3 and bar 5) differs. The authors should explain this discrepancy.

Reply: Thank you for your comments. In Figure 6B, the relative luciferase activity of the two controls (bar 3 and bar 5) is different. Bar 3 is the control of Bar 4. This set of experiments was to confirm the effect of AR knockdown on PRKACA promoter activity. However, Bar 5 is a control of Bar 6. This set of experiments was used to detect PRKACA promoter activity co-regulated by AR and CARM1. 

  1. Although ChIP experiments are depicted in Figure 6D, the manuscript lacks a detailed description of the ChIP protocol. It is recommended to include this information for transparency.

Reply: Sorry for that! The detailed description of ChIP protocol has been added to the Method section and marked in red. Please check!

  1. The manuscript does not mention the target sequences for the numerous shRNA constructs used. Authors should provide these sequences or, if commercially obtained, include catalog numbers for transparency.

Reply: Thank you for your comments. Lentiviruses LV-shRBM5, LV-shPRKACA and LV-shCARM1 used in this study were constructed by Shijiazhuang Biocaring Biotechnology Co., LTD. The shRNA lentiviruses are all based on the LV-2N (pGLVU6/Puro) skeleton vector (Catalog No.: BCR-shRNA-LV-2N). Relative comments have been added in the Method section and marker in red. Please check!

  1. In Section 3.7, the last sentence of results refers to Fig. 8, but according to the data, it should be Fig. 7E and F. Authors are advised to correct this reference for accuracy.

Reply: Thank you for your comments. In Section 3.7, the last sentence of the result referenced in Figure 8 is correct. In the penultimate sentence, the description of the results of Western blot is referred to figures 7E and 7F.

Reviewer 2 Report

Comments and Suggestions for Authors

The authors have attempted to demonstrate a regulatory mechanism of Wnt/β-catenin activation through the RBM5/CARM1/PRKACA axis and tried to identify a new potential target for the treatment of bladder cancer.

The study is methodologically well performed.

There are a few of points which may be considered for further improvement.

Please provide reference justifying Our previous study revealed that RBM5, as an important RBP member, was downregulated in BC tissues and cells. RBM5 OE effectively reduced the activation of the Wnt/β-catenin axis, which resulted in a decrease in cell proliferation.” Page 15

You stated in the discussion section „Previous studies have demonstrated that RBPs play essential roles in the tumorigenesis of BC cells“, and you only provided one reference, please provide all references that relate to this claim.

Please list the limitations of this study at the end of the discussion section.

The findings of this study could be of importance for clinical practice.

Author Response

Point-by-point response to referees

Reviewer 2#

 The authors have attempted to demonstrate a regulatory mechanism of Wnt/β-catenin activation through the RBM5/CARM1/PRKACA axis and tried to identify a new potential target for the treatment of bladder cancer. The study is methodologically well performed. There are a few of points which may be considered for further improvement.

  1. Please provide reference justifying„Our previous study revealed that RBM5, as an important RBP member, was downregulated in BC tissues and cells. RBM5 OE effectively reduced the activation of the Wnt/β-catenin axis, which resulted in a decrease in cell proliferation.” Page 15

Reply: Thank you for your comments and suggestions. References are added to these sentences in the Discussion section and marked in red. Please check!

  1. You stated in the discussion section „Previous studies have demonstrated that RBPs play essential roles in the tumorigenesis of BC cells“, and you only provided one reference, please provide all references that relate to this claim.

 Reply: Thank you for your comments and suggestions. In the Discussion section, more references are added to these sentences and marked in red. Please check!

  1. Please list the limitations of this study at the end of the discussion section.

Reply: Thank you for your suggestions. Limitations of this study: 1) More clinical samples and more reliable animal experiments, such as patient-derivedxenografts (PDX), are needed to enhance the persuasive results. 2) Whether the relationship between AR and RBM5 has a feedback loop regulation mode. The above problems need further in-depth study. These comments have added to the Discussion section and marked in red.

  1. The findings of this study could be of importance for clinical practice.

 Reply: Thank you for your comments.

Reviewer 3 Report

Comments and Suggestions for Authors

The work clearly describes a relevant pathway involved in carcinogenesis of bladder cancer and is adequately structured. Just minor English revision is needed.

Comments on the Quality of English Language

Conjugation of some verbs in abstract and resultas shoukld be revised.

Author Response

Point-by-point response to referees

Reviewer 3#

Comments and Suggestions for Authors

  1. The work clearly describes a relevant pathway involved in carcinogenesis of bladder cancer and is adequately structured. Just minor English revision is needed.

 Reply: Thank you for your comments. This manuscript has been edited by native English speakers. Thank you.

Reviewer 4 Report

Comments and Suggestions for Authors

cancers-2751491

Zhang et al. demonstrated that the downregulation of RBM5 promoted the expression of CARM1 in BC cells and tissues. They also showed that increase of CARM1 activated the Wnt/β-catenin axis and cell proliferation, which then contributed to the poor prognosis of patients with Bladder Cancer. They found that RBM5 bound directly to CARM1 mRNA and promoted exon 9 skipping in AS-NMD. They also revealed that protein kinase catalytic subunit alpha (PRKACA) was regulated by CARM1 at the transcription level, and promoted the growth and progression of BC cells.

The results are potentially interesting, but there are several concerns for this manuscript.

1) In Figure 4, the authors used Dox for inhibition of RNA synthesis. However, Dox inhibit both DNA polymerase and RNA polymerase. They should use either alpha-amanitine or Actinomycin D for this purpose.

2) In the same Figure, the authors describe RBM5 binds to CDS of CARM1. Where is the binding site of RBM5 in CARM1 mRNA? If RBM5 regulates CARM1 alternative splicing, it is likely that RBM5 binds either exon or intron element of RBM5 pre-mRNA. The authors should try whether RBM5 can precipitate CARM1 pre-mRNA or not.

3) What is the mechanism for RBM5 to promote exon 9 skipping in AS-NMD?

4) In Figure 8, the authors draw ellipse. If this means the nucleus, RBM5 should be in the nucleus. In addition, RBM5 functions in alternative splicing, not protein expression/modification/stability directly. Which results demonstrate CARM1 directly promotes transcription of PRKACA together with AR? Does AR mean androgen receptor?

Comments on the Quality of English Language

English writing is fine, but English editing is recommended.

Author Response

Point-by-point response to referees

Reviewer 4

Zhang et al. demonstrated that the downregulation of RBM5 promoted the expression of CARM1 in BC cells and tissues. They also showed that increase of CARM1 activated the Wnt/β-catenin axis and cell proliferation, which then contributed to the poor prognosis of patients with Bladder Cancer. They found that RBM5 bound directly to CARM1 mRNA and promoted exon 9 skipping in AS-NMD. They also revealed that protein kinase catalytic subunit alpha (PRKACA) was regulated by CARM1 at the transcription level, and promoted the growth and progression of BC cells.

The results are potentially interesting, but there are several concerns for this manuscript.

1) In Figure 4, the authors used Dox for inhibition of RNA synthesis. However, Dox inhibit both DNA polymerase and RNA polymerase. They should use either alpha-amanitine or Actinomycin D for this purpose.

 Reply: Thank you for your comments and suggestions. First of all, you are right about the use of doxycycline (Dox) here. The purpose of Dox here is to inhibit new RNA synthesis. At the same time, the degradation rate of the synthesized RNA in the cell was detected, that is, the participation of RBM5 in the metabolism of CARM1 RNA was observed. Therefore, the function of Dox is the same as that of alpha-amanitine or Actinomycin D. In addition, experiments like this use the same Dox 2. So I think using Dox is feasible and compelling. Maybe in the next step, we try to make alpha-amanitine or Actinomycin D and see how they differ from using Dox. Thank you!

2) In the same Figure, the authors describe RBM5 binds to CDS of CARM1. Where is the binding site of RBM5 in CARM1 mRNA? If RBM5 regulates CARM1 alternative splicing, it is likely that RBM5 binds either exon or intron element of RBM5 pre-mRNA. The authors should try whether RBM5 can precipitate CARM1 pre-mRNA or not.

Reply: Thank you for your comments and suggestions. Your understanding is deep and correct. Combined with previous reports and our experimental results, the main binding motif of RBM5 is GAAGGAA or GAAGGAG 3,4. However, there was a GAAGGAG motif in exon 9 (86 nt) of CARM1 mRNA. Although there is also a GAAGGAG motif in the first intron, RBM5 may not be functioning AS an AS-NMD due to the length of the intron (33028 nt) (too far from the splice recognition site). We performed RNA co-immunoprecipitation with RBM5 antibody, and the pre-mRNA of CARM1 could not be detected.

3) What is the mechanism for RBM5 to promote exon 9 skipping in AS-NMD?

Reply: Thank you for your comments. Alternative splicing (AS) is largely regulated by interactions between cis-regulatory elements in pre-mRNA and transacting splicing regulators, which are primarily RNA binding proteins (RBPs). AS also produces mRNA variants that are substrates for nonsense-mediated mRNA decay (NMD). AS events that generate premature termination codons (PTCs) are often coupled with NMD, which is important not only for eliminating aberrant mRNA transcripts containing PTCs, but also for post-transcriptional tuning of gene expression. The mRNA transcripts containing a PTC located more than 50–55 nt upstream from the last exon-exon junction or a long 3’-untranslated region (3’UTR) are often substrates for NMD. mRNA degradation by this pathway depends on translation and subsequent recruitment of essential NMD factors, including the central factor, up frameshift 1 (UPF1). RBM5 may promote exon 9 skipping by binding to exon 9 of the CARM1 pre-mRNA and by recruiting essential NMD factors such as frameshift 1 (UPF1).

4) In Figure 8, the authors draw ellipse. If this means the nucleus, RBM5 should be in the nucleus. In addition, RBM5 functions in alternative splicing, not protein expression/modification/stability directly. Which results demonstrate CARM1 directly promotes transcription of PRKACA together with AR? Does AR mean androgen receptor? 

Reply: Thank you for your comments. The schematic diagram in Fig. 8 has been revised. RBM5 in the nucleus inhibited CARM1 expression by alternative splicing. The results of Fig. 6A to 6D demonstrate CARM1 directly promotes transcription of PRKACA together with androgen receptor (AR). Please check!

1 Zhang, Y. P. et al. Down-regulated RBM5 inhibits bladder cancer cell apoptosis by initiating an miR-432-5p/beta-catenin feedback loop. FASEB J 33, 10973-10985 (2019). https://doi.org/10.1096/fj.201900537R

2 Sun, Y. et al. Autoregulation of RBM10 and cross-regulation of RBM10/RBM5 via alternative splicing-coupled nonsense-mediated decay. Nucleic Acids Res 45, 8524-8540 (2017). https://doi.org/10.1093/nar/gkx508

3 Soni, K. et al. Structural basis for specific RNA recognition by the alternative splicing factor RBM5. Nat Commun 14, 4233 (2023). https://doi.org/10.1038/s41467-023-39961-w

4 Mourao, A. et al. Structural basis for the recognition of spliceosomal SmN/B/B' proteins by the RBM5 OCRE domain in splicing regulation. Elife 5 (2016). https://doi.org/10.7554/eLife.14707

Round 2

Reviewer 4 Report

Comments and Suggestions for Authors

It seems to me that the authors at least tried their best to revise the manuscript.

Comments on the Quality of English Language

English writing is OK.

Round 3

Reviewer 4 Report

Comments and Suggestions for Authors

I have no more comments and concerns for this manuscript.